biomaterials/biochemistry/biomedical engineering

PAMAM-COOH, collagen fibrils, dentine, matrix metalloproteinases, enzymology, adhesive

**Author for correspondence:**
Lisha Gu
e-mail: gulisha@mail.sysu.edu.cn

# The inhibitory effect of carboxyl-terminated polyamidoamine dendrimers on dentine host-derived matrix metalloproteinases *in vitro* in an etch-and-rinse adhesive system

Qian Wu, Tiantian Shan, Manduo Zhao, Sui Mai and Lisha Gu

Department of Endodontics, Guangdong Provincial Key Laboratory of Stomatology, Guanghua School of Stomatology, Hospital of Stomatology, Sun Yat-Sen University, Guangzhou 510055, Guangdong, People's Republic of China

QW, 0000-0002-0625-1543; LG, 0000-0002-5655-3021

The biomimetic remineralization of collagen fibrils has increased interest in restoring the demineralized dentine generated by dental caries. Carboxyl-terminated polyamidoamine dendrimers (PAMAM-COOH), hyperbranched polymeric macromolecules, can act as non-collagenous proteins to induce biomimetic remineralization on a dentine organic matrix. However, *in vivo* remineralization is an extremely time-consuming process; before complete remineralization, demineralized dentine collagen fibrils are susceptible to degradation by host-derived matrix metalloproteinases (MMPs). Therefore, we examined the effect of fourth-generation PAMAM-COOH (G4-PAMAM-COOH) on the collagenolytic activities of endogenous MMPs, the interaction between G4-PAMAM-COOH and demineralized dentine collagen and the influence of G4-PAMAM-COOH pre-treatment on resin–dentine bonding. G4-PAMAN-COOH not only inhibited exogenous soluble rhMMP9 but also hampered the proteolytic activities of dentine collagen-bound MMPs. Cooperated with the results of G4-PAMAM-COOH absorption and desorption, FTIR spectroscopy provided evidence for the exclusive electrostatic interaction rather than hydrogen or covalent bonding between G4-PAMAM-COOH and dentine collagen. Furthermore, G4-PAMAM-COOH pre-treatment showed no

damage to resin–dentine bonding because it did not significantly decrease the elastic modulus of the demineralized dentine, degree of conversion, penetration of the adhesive into the dentinal tubules or ultimate tensile strength. Thus, G4-PAMAM-COOH can effectively inactivate MMPs, retard the enzymolysis of collagen by MMPs and scarcely influence the application of resin–dentine bonding.

## 1. Introduction

Resin adhesive restoration is a popular preferred procedure for repairing dental tissue defects due to caries and uses an etchant or acidic monomers to expose collagen fibres for the purpose of permitting adhesive resin monomers to infiltrate dentine and then form a resin–dentine hybrid layer (HL). However, due to incomplete resin infiltration, the mineral-depleted, resin-sparse, water-rich collagen at the bottom of the HL is susceptible to deterioration during prolonged function, adversely affecting the durability of the resin–dentine bonding. It has been reported that collagen-bound matrix metalloproteinases (MMPs) in human dentine are responsible for the degradation of partially non-impregnated collagen fibrils [1–3]. MMPs exposed by dentine demineralization procedures can be activated on the account of acidic monomers in etch-and-rinse or self-etch adhesive systems [1,4,5].

Currently, several approaches can be used to improve the durability of resin–dentine bonding. The use of synthetic and natural cross-linkers has been confirmed to increase the mechanical performance of dentine [6–8] by facilitating additional collagen cross-linking [9–11] and inactivating endogenous dentine MMPs [12,13] mainly by concealing the cleavage sites of collagen [14] and/or altering the spatial structure of proteases, resulting in enhanced resistance against the enzymic degradation of dentine matrix. MMPs inhibitors, such as chlorhexidine, quaternary ammonium salts and zinc, are applied to suppress collagenolytic activity in the dentine–collagen matrix due to changes in protein configuration or to disturb the protease-activating process [5,15–17]. Similar to collagen cross-linkers, the main drawback of non-specific inhibitors to inactivate MMPs and cysteine cathepsins is the residue of the water-rich, resin-sparse and mineral-depleted collagen matrix within the HL. Although integrity of the denuded collagen fibrils can be preserved, the flaccid collagen matrices are susceptible to creep or cyclic fatigue rupture after prolonged function [18]. The deposition of apatite crystals in intrafibrillar and interfibrillar compartments is crucial for maintaining and stabilizing the mechanical properties of the dentine collagen [19]. Data reported in the literature suggested that biomimetic remineralization can restore water-filled, resin-sparse regions within the HL to intact mineralized dentine characterized by excellent mechanical properties and fossilize endogenous collagenolytic enzymes [20–22], resulting in enhanced resistance of collagen against fatigue rupture and degradation. Therefore, the biomimetic remineralization of demineralized dentine collagen beneath the HL has been proposed as a method superior to the uses of cross-linkers and MMP inhibitors. The implementation of biomimetic remineralization is closely dependent on three elements, involving non-collagenous proteins (NCPs), collagen scaffold and extraneous minerals (calcium and phosphorus).

NCPs are essential for the regulation of tissue mineralization by stabilizing metastable amorphous calcium phosphate (ACP) precursors and orchestrating the alignment of ACP precursors during their transformation into apatite [23,24]. In view of the commercial unavailability of native or recombinant NCPs, scientists have resorted to the use of NCP analogues to mimic functional domains of naturally occurring proteins [23,25]. Polyamidoamine dendrimers (PAMAM) are a new kind of hyperbranched polymeric macromolecules with well-defined dimensions and low cytotoxicity [26]. The structure of PAMAM can be classified into three components: the core, the interior and the shell [27]. The interior is composed of repetitive branching units, which dominantly affects the morphology of PAMAM. The more branching units it has, the higher generation PAMAM is defined as. The surface of PAMAM can be modified with functional peripheric radicals, introducing different surface charges. Several studies have reported the ability of positively, neutrally and negatively charged dendrimers to interact with anionically and/or cationically charged proteins [28–30]. Furthermore, it has been reported that PAMAM possesses the properties of predominant biomimetic analogues [31–33], especially carboxyl-terminated PAMAM (PAMAM-COOH) dendrimers. Compared with the previously reported NCPs analogues such as polyacrylic acid and sodium tripolyphosphate, PAMAM-COOH as a multifunctional NCPs analogue can modulate highly ordered intrafibrillar mineralization on the organic dentine matrix [34]. Additionally, collagen scaffold, as the template for hydroxyapatite deposition, plays an important role in biomimetic remineralization. However, *in vivo* remineralization is an extremely time-consuming process; during remineralization, demineralized dentine collagen

fibrils exposed by etching procedure may be susceptible to degradation by MMPs, resulting in unsatisfactory remineralization. Hence, the degradation of collagen fibrils by MMPs has become the main obstacle to obtaining satisfactory remineralization [35,36].

The objective of the current study was to explore whether G4-PAMAM-COOH can inhibit endogenous MMPs in the dentine matrix. We also attempted to ascertain the interaction mechanism between G4-PAMAM-COOH and dentine collagen. Furthermore, the effect of G4-PAMAM-COOH on resin–dentine bonding was examined. The null hypotheses tested are as follows: (i) G4-PAMAM-COOH has no inhibitory effect on endogenous MMPs; (ii) G4-PAMAM-COOH has no negative influence on the properties of resin adhesive restoration.

# 2. Material and methods

## 2.1. Analysis of the effect of G4-PAMAM-COOH on MMPs

### 2.1.1. Inhibition of soluble rhMMP9

The effect of G4-PAMAM-COOH on soluble rhMMP9 was detected using purified recombinant human (rh) MMP9 (Catalogue no. 7789-10, BioVision, CA, USA) and the SensoLyte Generic MMP assay kit (AnaSpec Inc., Fremont, CA, USA). When the substrate provided by the assay kit is disintegrated by MMPs, a coloured product can be released gradually. The intensity of the colour is dynamically metabolic during incubation and can be detected by a microplate reader at 412 nm.

G4-PAMAM-COOH (CYD-140, Chenyuan, Weihai, Shandong, China) was dissolved in deionized water to obtain the required concentrations. A constant concentration of rhMMP9 (10 ng ml$^{-1}$) was acquired by mingling 10 µg rhMMP9 and 1 ml PBS buffer (pH: 7.4). Triplicate examinations were executed for each group [37]. The test compounds involved various concentrations of G4-PAMAM-COOH (10, 5, 2.5, 1, 0.5, 0.25 and 0.125 mg ml$^{-1}$). The inhibitor (GM6001) from the assay kit was used as the 'inhibitor control', and the 'positive control' was composed of the substrate and rhMMP9 without the test compounds. The usage in each group complied with the manufacturer's instructions. In accordance with the previously measured standard curve, the final absorbance was converted to relevant concentration. The potency of rhMMP9 inhibition/promotion by G4-PAMAM-COOH was manifested as percentages of the acquired 'positive control' concentration, which was treated as the maximum concentration.

### 2.1.2 In situ zymography

Twenty freshly extracted non-carious human third molars were obtained from the Department of Maxillofacial Surgery in the Oral Hospital Affiliated to Guanghua College of Stomatology following the patients' informed consent for the use of teeth for the experiments. The teeth were stored in 0.9% (w/v) NaCl at 4°C and used timely within one week after they were extracted. One millimetre thick slices from the middle coronal dentine were obtained from each tooth by a slow-speed saw (Accutom-50, Struers, UK) under water cooling. Six-hundred-grit wet silicon-carbide paper was used to create a standard smear layer on the coronal surface of each disc, and dentine was etched for 15 s with a 35% phosphoric acid gel (3 M ESPE, St Paul, MN, USA) and rinsed with a water–air spray for 30 s to remove the residual acid gel and soluble mineral. According to a preliminary experiment, 8 mg ml$^{-1}$ was an optimum concentration for adsorption to dentine collagen. Therefore, this concentration was employed in the following experiments. Eight milligrams G4-PAMAM-COOH were dissolved with 1 ml deionized water as primer.

The etched dentine specimens were randomly divided into two different groups ($n = 10$). The experimental group was pre-treated with 8 mg ml$^{-1}$ G4-PAMAM-COOH for 1 min; the control group was pre-treated with deionized water for 1 min. And then every sample was rinsed with deionized water for 5 s. Excess water was blotted with absorbent paper when the dentine surface was still moist. The bonding procedure was performed with Single Bond 2 adhesive (3 M ESPE) in accordance with the manufacturer's instructions. A 1 mm thick flowable composite (Filtek Flow; 3 M ESPE) was applied to the bonded discs and light-cured for 20 s using a quartz–tungsten–halogen light-curing unit (Curing Light 2500; 3 M ESPE). After bonding, the dentine discs from each group were divided into two subgroups ($n = 5$). One subgroup was stored in deionized water for 24 h at 37°C, labelled as T0. The other subgroup was exposed to thermal cycling (10 000 thermal cycles, 5 and 55°C for 60 s) [38] to simulate 1 year of intraoral use, labelled as T1. Each bonded specimen was sectioned vertically into 1 mm thick slabs to expose the resin–dentine interface using a slow-speed saw. Each bonded slab was affixed to a glass slide with cyanoacrylate cement, and an approximately 50 µm thick section was

**Table 1.** Groups used for measuring released ICTP. The total volume of the experimental groups was 1000 µl, leading to a two-fold dilution of G4-PAMAM-COOH. Therefore, the final concentration of G4-PAMAM-COOH was 8 mg ml$^{-1}$.

| group | test solution and volume | final concentration of G4-PAMAM-COOH |
|---|---|---|
| 1 | 1000 µl deionized water | 0 mg ml$^{-1}$ |
| 2 | 500 µl artificial saliva, 500 µl deionized water | 0 mg ml$^{-1}$ |
| 3 | 500 µl 16 mg ml$^{-1}$ G4-PAMAM-COOH (dissolved in deionized water) 500 µl deionized water | 8 mg ml$^{-1}$ |
| 4 | 500 µl 16 mg ml$^{-1}$ G4-PAMAM-COOH (dissolved in deionized water) 500 µl artificial saliva | 8 mg ml$^{-1}$ |

obtained by serially polishing with 500- and 1200-grit wet silicon-carbide papers. A smooth and slippery surface was achieved by further polishing with 2000-, 2500- and 4000-grit wet silicon-carbide papers [39]. An *in situ* zymography protocol reported by Mazzoni *et al*. was performed [40]. Fifty microlitres quenched fluorescein-conjugated gelatin (E-12055; Molecular Probes, Eugene, OR, USA) were used as an MMP substrate and were pipetted onto each slab, which was then covered with a coverslip. The slides were kept away from light and incubated in a 100% relative humidity chamber at 37°C for 48 h. The hydrolysis of the quenched fluorescein-conjugated gelatin, which is indicative of endogenous gelatinolytic activity, was evaluated with a confocal laser scanning microscope (CLSM; excitation/emission: 488/530 nm; LSM780; Carl Zeiss). The image acquisition was in accordance with the method in a previous paper of our research group [39]. The quantitative analysis of fluorescein intensity was performed using ImageJ software.

### 2.1.3. Measurement of released ICTP

Type I collagen degradation by MMPs may release C-terminal cross-linked telopeptides (ICTP), which are regarded as a marker of MMP activity. Fifty non-carious, freshly extracted human third molars were selected, and two parallel sections were performed perpendicularly to the longitudinal axis of the tooth to remove the occlusal enamel and root. The pulpal soft tissues were obtained from the exposed pulp chamber. The residual enamel and cementum attached to the crown periphery were removed using a high-speed hand saw with constant water cooling. The resulting crowns were placed in a stainless steel screw-top tube with a stainless steel ball, frozen in liquid nitrogen for 3 min and triturated at 40 Hz for 3 min in a mill (JXFS TPRP-24, Jingxin, Shanghai, China). This treatment was repeated until the resulting powder passed through a designated sieve (bore diameter less than 40 µm). The dentine powder was completely demineralized in 10 wt% phosphoric acid (pH: 1.0) at 4°C for 8 h. The depletion of minerals in the powder was confirmed by digital radiography. The demineralized dentine powder was rinsed four times with deionized water and lyophilized. The powder was divided into four groups, and each group was divided into five 10 mg aliquots subpackaged in microcentrifuge tubes. Each 10 mg aliquot of powder was immersed in the corresponding test solution (table 1). The incubation medium was collected after two weeks of incubation and was used to quantify the collagenolysis of the demineralized dentine matrix using an ICTP ELISA kit (Catalogue no. CSB-E11224 h, Cusabio, Wuhan, China).

## 2.2. Analysis of interaction between dentine collagen and G4-PAMAM-COOH

### 2.2.1. Fourier transform infrared spectroscopy measurements

Demineralized dentine powder was obtained as depicted in the part on measuring released ICTP. Fourier transform infrared spectroscopy (FTIR) was performed (IS10, Nicolet, USA) in the spectral range of 500–4000 cm$^{-1}$ with a resolution of 4 cm$^{-1}$. Ten milligrams demineralized dentine powder were immersed in 1 ml G4-PAMAM-COOH (8 mg ml$^{-1}$) for 30 min. The mixture was separated by centrifugation. The residual liquid in the dentine powder was absorbed by absorbent paper. The dentine powder pellets were irrigated three times by deionized water and lyophilized. The anhydrous

G4-PAMAM-COOH and the dentine powder treated by G4-PAMAM-COOH (8 mg ml$^{-1}$) was also scanned by FTIR.

### 2.2.2. G4-PAMAM-COOH adsorption

Dentine powder was prepared as described above. The demineralized dentine powder was resuspended in 30 ml of deionized water to flush away all traces of the reaction products. The background absorbance was reduced to 0.09 absorbance units at 228 nm (EpochTM, BioTek Instruments, Inc.) by washing the powder with deionized water every 8 h for 15 days [17]. Then, the powder was immediately used for the G4-PAMAM-COOH adsorption experiment.

Following the protocol of Blackburn *et al.* [41], 35 portions of dentine power (100 mg each) were distributed into seven groups ($n = 5$) and immersed in 1 ml of 1.25, 2.5, 5, 6, 8, 10 or 12 mg ml$^{-1}$ standard G4-PAMAM-COOH solution for 30 min. Then, the microcentrifuge tubes containing the previous mixture were centrifuged at 14 000 r.p.m. for 5 min. Five hundred microlitres supernatant were transferred into another microcentrifuge tube and centrifuged again. Two hundred microlitres supernatant from the surface layer were pipetted into an ultraviolet–visible (UV)-transparent 96-well plate, and the absorbance was measured in duplicate at 228 nm using an ultraviolet spectrophotometer. The absorbance was converted to G4-PAMAM-COOH concentration according to the standard curve. The G4-PAMAM-COOH concentration of the supernatant was less than that of the standard solution due to the addition of the demineralized dentine powder, revealing G4-PAMAM-COOH uptake by the dentine powder. This value was labelled as the PAMAM uptake at the endpoint. The absorption of G4-PAMAM-COOH by the demineralized dentine powder was calculated using the following equation

$$PAMAM_{uptake} = \frac{PAMAM_{STD} - PAMAM_{end}}{g \ dry \ wt},$$

where $PAMAM_{STD}$ is the G4-PAMAM-COOH standard concentration without powder (mg ml$^{-1}$) and $PAMAM_{end}$ is the endpoint G4-PAMAM-COOH concentration in solution (mg ml$^{-1}$) after mixing with the powder, divided by the weight of the dry dentine powder.

### 2.2.3. G4-PAMAM-COOH desorption

The dentine powder was separated from the supernatant, and a strip of dry paper was used to absorb the excess remaining solution. The powder was immersed in 1 ml of deionized water or 0.5 M NaCl. Water can extract materials without bonding to the substrate. Sodium chloride can compete with electrostatically bound materials, resulting in the desorption of bound materials from the substrate [42]. The mixture was sonicated for 15 min and then centrifuged at 14 000 r.p.m. for 5 min. Four hundred microlitres supernatant were centrifuged again, and then 200 µl supernatant was transferred into a UV-transparent 96-well plate and measured at 228 nm by an ultraviolet spectrophotometer.

## 2.3. Assessment of the impact of G4-PAMAM-COOH on resin–dentine bonding

### 2.3.1. Nanoindentation by atomic force microscopy

Three millimetre thick dentine beams ($n = 20$) were sectioned from the mid-coronal dentine using a low-speed saw under water cooling and demineralized in 10 wt% H$_3$PO$_4$ for 24 h at 20°C until demineralization was complete. The samples were immersed in deionized water for 2 h at 4°C. One half of the lyophilized samples ($n = 10$) were treated by deionized water, and the other half ($n = 10$) were immersed in 8 mg ml$^{-1}$ G4-PAMAM-COOH until resilience was recovered. The elasticity modulus of the specimens was tested using atomic force microscopy (AFM, Dimension FastScan, Bruker, USA). The AFM probe used was SNL-A ($k = 0.48$ N m$^{-1}$) with a maximum loading of 5 nN; Poisson's ratio of the sample was 0.5. The elasticity modulus of the specimens was measured by an analysis software (NanoScope Analysis, v. 1.4, Bruker).

### 2.3.2. Degree of conversion

Ten freshly extracted, non-carious human third molars were prepared as described in *In situ* zymography. The teeth were divided into two groups. The experimental group was preconditioned with 8 mg ml$^{-1}$ G4-PAMAM-COOH for 1 min after etching with the 35% phosphoric acid gel, and the

control group was pre-treated with deionized water. The resulting resin–dentine discs were sectioned into several slices (each 1 mm thick). Two slices in the middle of each disc were selected to analyse the degree of conversion (DC). The DC of each group ($n = 6$) was assessed by micro-Raman spectroscopy at 785 nm wavelength (Renishaw inVia, UK). A microscope (Leica DM/LM optical microscope; Leica, Wetzlar, Germany) was connected to micro-Raman spectroscopy. With the help of the microscope, a laser beam generating an approximately 1 µm spot focused on the intertubular regions of the resin–dentine interface. The spectra were acquired in line scans and calculated using the following formula [43]:

$$DC\% = \left(1 - \frac{R_{\text{cured}}}{R_{\text{uncured}}}\right) \times 100\%,$$

where $R$ represents the ratio between the peak intensities at 1640 cm$^{-1}$ and at 1610 cm$^{-1}$ of the light-cured adhesive at the resin–dentine interface and uncured adhesive within the quartz capillary siphon, respectively.

### 2.3.3. Adhesive permeation

Twenty non-carious, freshly extracted human third molars were selected and sectioned perpendicularly to the longitudinal axis of the tooth 3 mm below the cement-enamel junction using a slow-speed saw under water cooling, to remove the roots. A second parallel cut removed the occlusal enamel and superficial dentine to create a flat dentine surface in the middle of the crown, and 2.5 mm thick dentine was maintained above the pulpal horns using a calliper. The pulpal tissue was gently extracted from the exposed pulp chamber with a probe, avoiding pre-dentine damage. The crown sections were adhered to fenestrated Perspex discs and linked to polyethylene tubes to simulate pulpal pressure [44]. A 0.1% fluorescein sodium (Sigma-Aldrich, St Louis, MO, USA) aqueous solution was flowed through the tubes and infused into the pulpal chambers using a syringe, and 10 cm of water pressure was maintained for 48 h to ensure that the dentinal tubules and collateral branch were filled with fluorescein dye. After 48 h of pressure perfusion, the teeth were divided into two groups (10 teeth per group). Each group was further divided into two subgroups ($n = 5$) according to a different simulated pulpal pressure (0 or 5 cm of water pressure) generated by the 0.1% fluorescein sodium aqueous solution. A water pressure of 0 or 5 cm was maintained during the acid-etching process, pre-treatment, adhesive procedures and resin build-up. The exposed dentine surface was etched for 15 s with the 35% phosphoric acid gel and rinsed with continuous water irrigation for 30 s. After etching and rinsing, the experimental group was pre-treated with 8 mg ml$^{-1}$ G4-PAMAM-COOH for 5 min, and excess liquid was blotted with absorbent paper. The same procedure was applied to the control group with distilled water. Three drops of Single Bond 2 adhesive were mixed with tetramethylrhodamine B isothiocyanate (Sigma-Aldrich), forming a homogeneous solution [45]. The dyed adhesive was employed referring to the manufacturer's introduction. A 2 mm thick composite resin (Filtek Z350; 3 M ESPE) was constructed on the dentine in two sequential increments of 1 mm, each light-cured for 20 s. Each bonded tooth was prepared using the previously described *in situ* zymography method. After an approximately 50 µm thick section was acquired, a 20 µl drop of anti-fading agent (Mounting Medium with DAPI H-1200, Vectashield, Vector Laboratories Ltd, Cambridgeshire, UK) was placed onto the surface of each section and covered with a coverslip. Fluorescence images of the adhesive interface were generated by a confocal laser scanning microscope (×40 oil, Zoom 2.0). Three continuous areas of each specimen, located at the middle area of the two enamel–dentine junctions, were acquired for quantitative evaluation. The quantitative evaluation of the dyed adhesive permeation was performed using ImageJ software and is presented as a percentage of red fluorescence within the dentinal tubules. For each specimen, the adhesive penetration mean percentage of three regions was reserved for statistical analysis.

### 2.3.4. Ultimate tensile strength testing

Forty freshly extracted non-carious human third molars were divided into two groups and prepared, as described in *In situ* zymography. After pre-treatment and bonding, a 4 mm thick composite resin was built up on the dentine in four sequential increments of 1 mm, and each was light-cured for 20 s. All teeth were cut perpendicularly to the bonded interface into beams obtaining a number of I-shaped beam specimens (0.8 × 0.8 × 7 mm). Then, resin–dentine beams from each group were distributed into two subgroups ($n = 10$). One subgroup was stored in deionized water for 24 h at 37°C and labelled *T*0. The other subgroup was exposed to thermal cycling (10 000 thermal cycles, 5 and 55°C for 60 s) and labelled *T*1. Then, each beam was affixed to a jig on a universal testing machine (AG-1; Shimadzu, Kyoto, Japan)

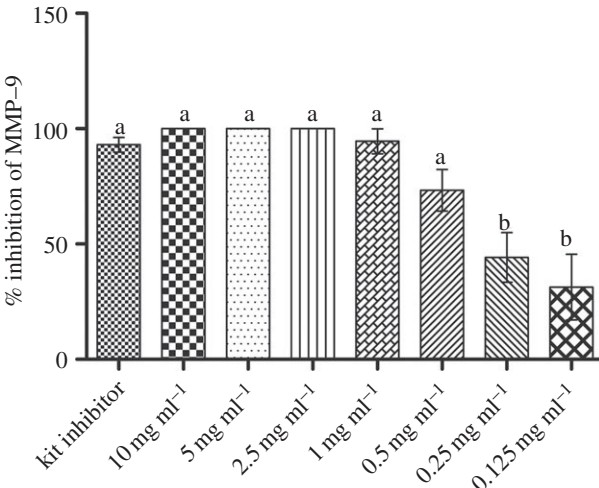

**Figure 1.** The per cent inhibition of rhMMP9 by different concentrations of PAMAM-COOH. The values are the means and standard deviations of the percentage. The groups labelled with the same letters on top of the bars are not significantly different ($p > 0.05$).

and subjected to a tensile force at a cross-head speed of 1 mm min$^{-1}$ until fracture [46]. The ultimate tensile strength (UTS) was measured by the following formula: UTS $= F/S$, where $F$ represents the failure force and $S$ is the cross-sectional area at the fracture site.

## 2.4. Statistical analysis

Since the normality and homoscedasticity assumptions of the data appeared to be valid, the percentage inhibition of MMP-9 in the eight groups and the amount of released ICTP in the four subgroups were analysed using one-way ANOVA and Tukey multiple comparison tests. The data in the section of nanoindentation and DC examinations were analysed by non-paired $t$-test. The data in the other parts could not be transformed into a normal distribution, and so the Kruskal–Wallis test followed by Dunn's multiple comparison tests was carried out using IBM SPSS v. 20 (NY, USA). The significance level for these tests was set at $\alpha = 0.05$.

# 3. Results

## 3.1. Inhibitory effect of G4-PAMAM-COOH on MMPs

### 3.1.1. Inhibition of soluble rhMMP9

The effect of G4-PAMAM-COOH on soluble rhMMP9 is shown in figure 1. The per cent inhibition of rhMMP9 by the control was 93.0% ± 6.4%. Over 90% soluble rhMMP9 was inhibited by different concentrations of G4-PAMAM-COOH, ranging from 1 to 10 mg ml$^{-1}$. The extent of inhibition increased in a dose-dependent manner ($p < 0.05$).

### 3.1.2. *In situ* zymography

Figure 2 represents the relative per cent areas of the HLs of the quenched fluorescein-conjugated gelatin showing hydrolysis activities in the four subgroups. The representative CLSM images of dentine conditioned with deionized water and G4-PAMAM-COOH are shown in figures 3 and 4. The super-imposition of the fluorescence images over the differential interference contrast (DIC) images are presented in figures 3c,f and 4c,f. Figures 3 and 4 present the green fluorescence within the HLs after 48 h of incubation (T0) and thermal cycling (T1), respectively. For the control specimens conditioned with deionized water, an intense green fluorescence was detected within the HLs, as shown in figures 3c and 4c, corresponding to the relative fluorescence intensities of 76.9 ± 2.6% and 96.2 ± 0.6%, respectively. By contrast, the HLs in the experimental groups pre-treated with the 8 mg ml$^{-1}$ G4-PAMAM-COOH manifested weak green fluorescence in figures 3f and 4f, reaching relative intensities of 42.2±2.0% and 50.3 ± 1.8%, respectively. The percentage values of green fluorescence in

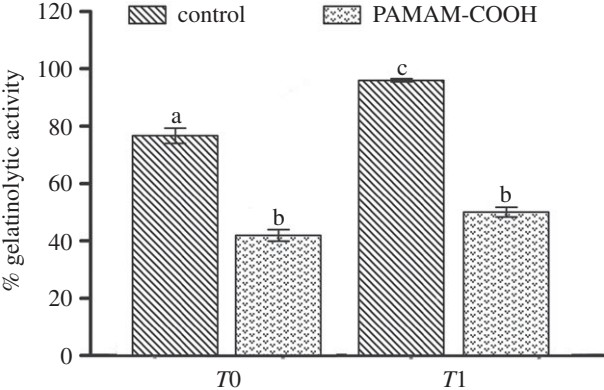

**Figure 2.** Mean and standard deviation of the relative fluorescence intensities of the hybrid layers in the four subgroups. T0, stored in deionized water for 1 day; T1, 10 000 thermal cycles. PAMAM-COOH, demineralized dentine interface conditioned with PAMAM-COOH; water (as control), demineralized dentine interface pre-treated with deionized water. For comparison of the four subgroups, the columns labelled with the different lowercase letters are significantly different ($p < 0.05$).

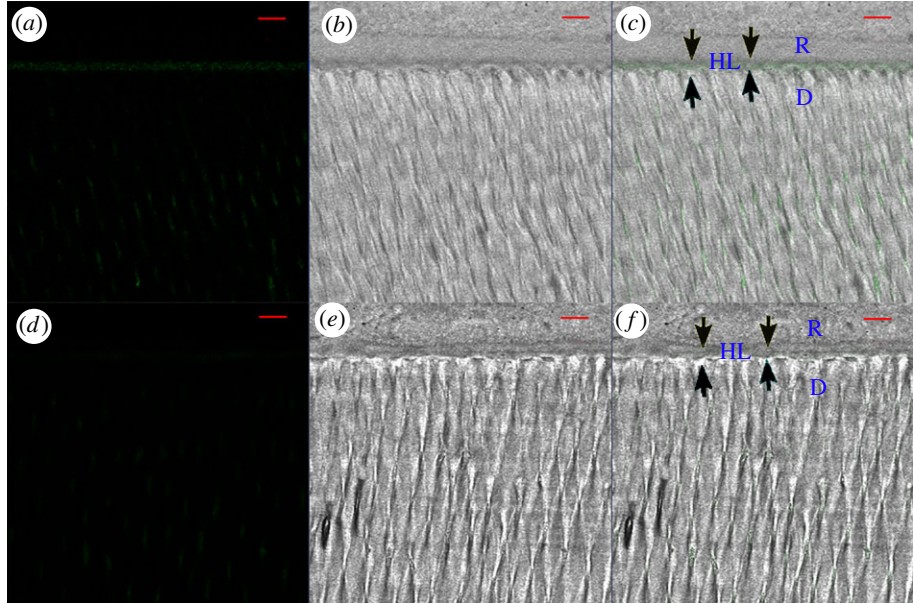

**Figure 3.** Resin–dentine interface pre-treated with deionized water (a–c) or 8 mg ml$^{-1}$ PAMAM-COOH (d–f) and bonded with Single Bond 2 adhesive after 24 h of storage (T0). The specimens were incubated for 48 h with quenched fluorescence-labelled gelatin. D, dentine; HL, hybrid layer (between arrowheads); R, adhesive resin; scale bar, 10 µm. (a) Confocal image in the green channel, showing fluorescence (indicating endogenous MMP activity) within the HL and the dentinal tubules. (b) DIC image showing the structure of the resin–dentine interface and dentinal tubules in water-conditioned specimen. (c) Merged image of (a,b). (d) Confocal image in the green channel, displaying fluorescence within the HL and the dentinal tubules. (e) DIC image showing the structure of the resin–dentine interface in the PAMAM-COOH-conditioned specimen. (f) Merged image of (d,e).

the experimental groups were significantly lower than those of the control groups irrespective of whether thermal cycling was performed ($p < 0.01$). With regard to the two time periods, the fluorescence intensity within the HLs significantly increased in the control group ($p < 0.05$) after thermal cycling (T1); however, the experimental group did not show significantly higher fluorescence intensity within the HLs at T1 than at T0 ($p = 0.839 > 0.05$)

### 3.1.3. Measurement of released ICTP

The descriptive statistics of ICTP telopeptides released from the demineralized dentine powder after two weeks of incubation are represented in figure 5. The release of the ICTP telopeptides was significantly

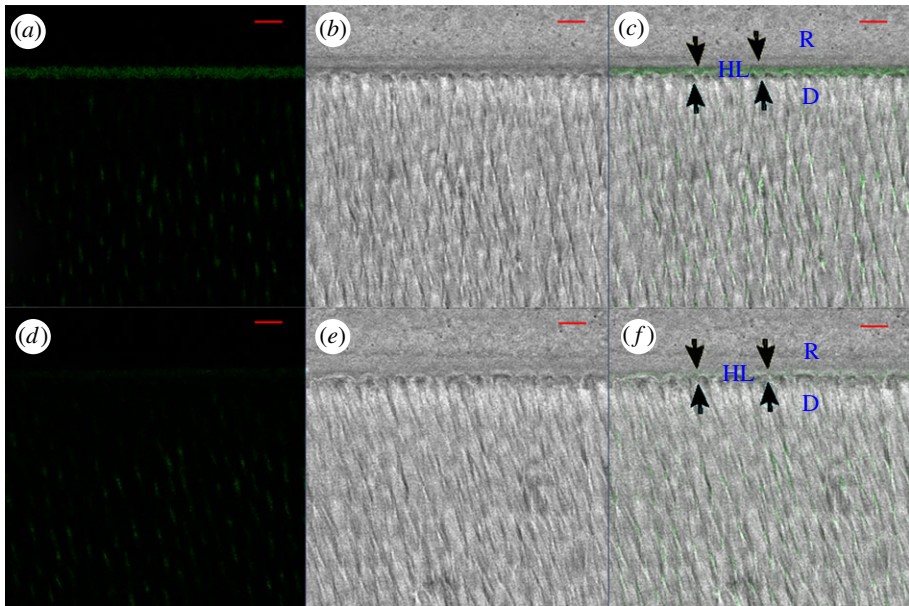

**Figure 4.** Resin–dentine interface pre-treated with deionized water (*a*–*c*) or 8 mg ml$^{-1}$ PAMAM-COOH (*d*–*f*) and bonded with Single Bond 2 adhesive after 10 000 thermal cycles (*T*1). The specimens were incubated for 48 h with quenched fluorescence-labelled gelatin. D, dentine; HL, hybrid layer (between arrowheads); R, adhesive resin; scale bar, 10 μm. (*a*) Confocal image in the green channel showing fluorescence (suggesting endogenous MMP activity) within the HL and the dentinal tubules. (*b*) DIC image showing the structure of the resin–dentine interface and dentinal tubules in the water-conditioned specimen. (*c*) Merged image of (*a,b*). (*d*) Confocal image in the green channel, showing fluorescence within the HL and the dentinal tubules. (*e*) DIC image showing the structure of the resin–dentine interface in the PAMAM-COOH-conditioned specimen. (*f*) Merged image of (*d,e*).

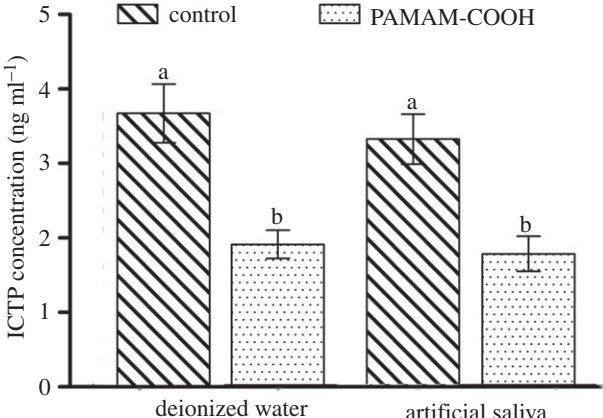

**Figure 5.** Mean and standard deviation of the concentration of ICTP released from demineralized dentine in four subgroups. Deionized water, the incubation solution without artificial saliva; Artificial saliva, the incubation solution containing artificial saliva; Control, demineralized dentine powder pre-treated with deionized water; PAMAM-COOH, demineralized dentine powder conditioned with PAMAM-COOH. For the comparison of the four subgroups, the columns labelled with the different lowercase letters are significantly different ($p < 0.05$).

affected by the pre-treatment ($p = 0.01 < 0.05$) and was not affected by the type of incubation medium ($p > 0.05$). In the control groups, the ICTP concentrations of the samples incubated in deionized water and artificial saliva were $3.67 \pm 0.39$ ng ml$^{-1}$ and $3.33 \pm 0.34$ ng ml$^{-1}$, respectively. However, the groups pre-treated with PAMAM-COOH showed a significantly lower release of ICTP telopeptides, reaching $1.90 \pm 0.19$ ng ml$^{-1}$ and $1.78 \pm 0.24$ ng ml$^{-1}$ for the samples incubated in deionized water and artificial saliva, respectively.

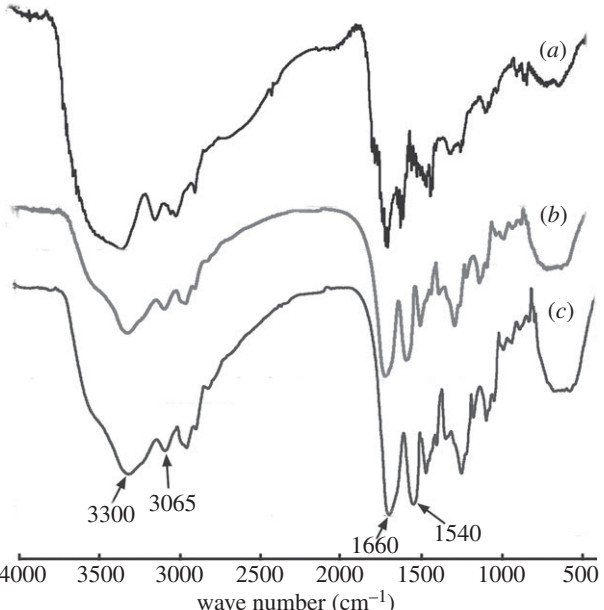

**Figure 6.** FTIR spectra of (*a*) PAMAM-COOH, (*b*) untreated dentine collagen and (*c*) dentine collagen treated by PAMAM-COOH.

## 3.2. Interaction between dentine collagen and G4-PAMAM-COOH

### 3.2.1. Fourier transform infrared spectroscopy

Figure 6 shows the corresponding FTIR spectra of G4-PAMAM-COOH, untreated dentine collagen and dentine collagen treated by G4-PAMAM-COOH. In the spectra of untreated (*b*) and G4-PAMAM-COOH-treated (*c*) dentine collagen, the amide A bands associated with N–H stretching vibration were observed at approximately 3300 cm$^{-1}$, and amide I bands were found at approximately 1660 cm$^{-1}$ indicating C=O stretching vibration conjugated with N–H bending vibration [47]. In addition, amide II bands related to both N–H bending and C–N stretching are presented at approximately 1540 cm$^{-1}$. The stretching vibrations described above were also observed in the spectrum of G4-PAMAM-COOH (figure 6*a*). It has been reported that a lower amide A frequency is attributed to the involvement of hydrogen bonds in the –NH$_2$ group of collagen [48]. The frequency of the amide A band did not shift to a lower wavelength after the G4-PAMAM-COOH treatment (figure 6*c*), indicating that there was no hydrogen bonding. The N–H stretching vibration and the C=O stretching vibration at approximately 1660 cm$^{-1}$ revealed –CONH– bonds. If the free –NH$_2$ groups in the collagen molecules reacted with the –COOH groups in the G4-PAMAM-COOH molecules, resulting in the formation of –CONH– bonds, the intensity of the amide I bands at approximately 1660 cm$^{-1}$ would increase [49]. An increase in the intensity ratio of the stretching vibration at approximately 1660 cm$^{-1}$ to approximately 3300 cm$^{-1}$ indicates increased amide linkages and covalent bonding of G4-PAMAM-COOH to dentine collagen. The intensity ratio of the stretching vibration at approximately 1660 cm$^{-1}$ to approximately 3300 cm$^{-1}$ of the demineralized dentine powder was $1.25 \pm 0.01$. The intensity ratio of the dentine powder treated by G4-PAMAM-COOH ($1.22 \pm 0.01$) was not significantly different from that of the untreated dentine powder ($p = 0.114 > 0.05$), which demonstrated that the interaction between G4-PAMAM-COOH and collagen was not dependent on covalent bonding.

### 3.2.2. G4-PAMAM-COOH adsorption and desorption

Figure 7 represents the amount of G4-PAMAM-COOH bound to the demineralized dentine powder after adsorption and desorption. In figure 7, curve a presents the amount of adsorbed G4-PAMAM-COOH that increased gradually in response to a higher concentration of G4-PAMAM-COOH and peaked at 8 mg ml$^{-1}$. However, there was a decline in the peak with rising concentration of G4-PAMAM-COOH. It has been reported that the PAMAM dendrimer self-aggregates with increasing concentration [29]. We speculated that the self-aggregation of dendrimers is responsible for the notable decline in absorption above 8 mg ml$^{-1}$. Therefore, the optimum concentration of G4-PAMAM-COOH adsorption to demineralized collagen was 8 mg ml$^{-1}$. Similar trends are also displayed in the deionized water

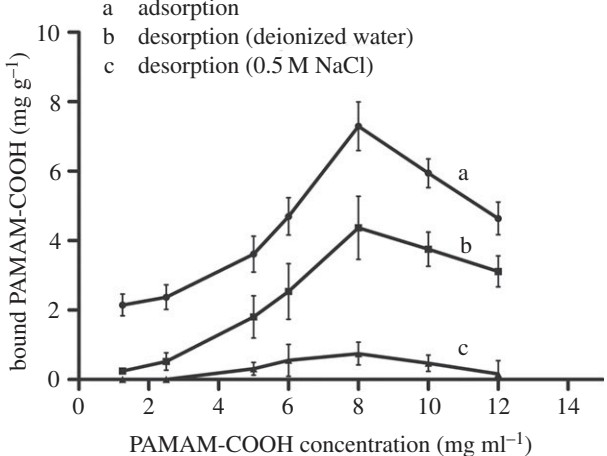

**Figure 7.** Adsorption and desorption of PAMAM-COOH by demineralized dentine powder. One hundred milligrams of powder were immersed in 1 ml of different standard concentrations of PAMAM-COOH. The three curves present the amounts of absorbed PAMAM-COOH after 30 min of absorption (curve a), water desorption (curve b) and 0.5 M NaCl desorption (curve c).

and 0.5 M NaCl desorption curves (in figure 7b,c). After desorption with deionized water, the amount of G4-PAMAM-COOH decreased compared to after adsorption (curve a). By contrast, curve c shows a dramatic decline in the amount of adsorbed G4-PAMAM-COOH, and the concentration of the residuum was close to 0 mg g$^{-1}$.

## 3.3. Assessment of the impact of G4-PAMAM-COOH on resin–dentine adhesive

### 3.3.1. Nanoindentation by AFM

The elastic modulus of the demineralized dentine discs immersed in deionized water and 8 mg ml$^{-1}$ G4-PAMAM-COOH were $4.06 \pm 0.45$ MPa and $4.72 \pm 0.57$ MPa, respectively. Statistical analysis showed that the elastic modulus of the dentine discs in the two groups was not significantly different ($p = 0.425 > 0.05$).

### 3.3.2. Degree of conversion

In the control group, the value of DC was $70.95\% \pm 0.90\%$; the value of DC in the experimental group reached $69.03 \pm 1.80\%$ and was not significantly different from that of the control group ($p = 0.362 > 0.05$).

### 3.3.3. Adhesive permeation

Representative CLSM images of the permeability of the resin–dentine interface created by the etch-and-rinse adhesive system under simulated pulpal pressures are shown in figures 8 and 9. The adhesive infiltration data that are expressed as the relative percentage of red adhesive at the site of the dentinal tubules are presented in figure 10.

In figures 8 and 9, the CLSM images illustrate the resin adhesive in red, and the positions of water penetration from the dentine substrate through the tubules in green. Figure 8 presents dense, branch-like resin tags at the resin–dentine interface and sufficient infiltration of the red resin adhesive into the dentinal tubules. In figure 9, droplet-type or short branch-type resin tags are observed, indicating relatively incomplete infiltration into the dentinal tubules. The shape and depth of the resin tags on the interface preconditioned by G4-PAMAM-COOH (figures 8e and 9e) are analogous to those in the control groups (figures 8b and 9b) under 0 or 5 cm of H$_2$O pressure. According to the quantitative analysis of permeation, the specimens under simulated pulpal pressure (0 cm of H$_2$O pressure) demonstrated significantly higher permeation into the dentinal tubules than the specimens under simulated pulpal pressure (5 cm of H$_2$O pressure) irrespective of whether G4-PAMAM-COOH or deionized water was applied ($p < 0.01$). In figure 10, the relative percentages of the permeability of the specimens pre-treated with deionized water under the two different simulated pulpal pressures were $27.3 \pm 6.2\%$ and $78.2 \pm 11.7\%$. The specimens conditioned with G4-PAMAM-COOH showed similar percentages of permeability, reaching $26.8 \pm 4.8$ and $71.9 \pm 12.0\%$. There was no significant difference

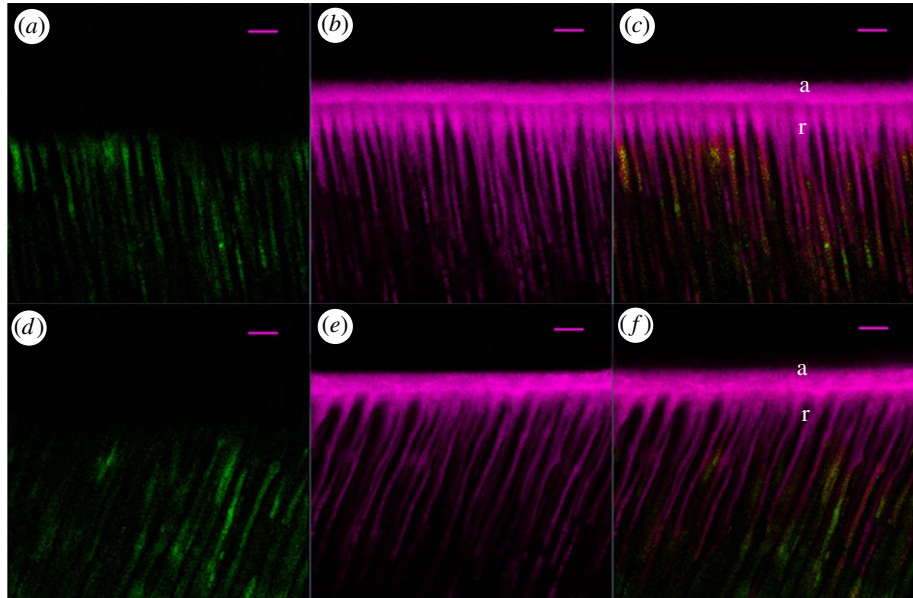

**Figure 8.** Representative CLSM fluorescence images of the resin–dentine interface created by the etch-and-rinse adhesive system without simulated pulpal pressure (0 cm of H$_2$O pressure) with or without PAMAM-COOH pre-treatment. The adhesive was dyed red, and water was dyed green. The red colour was adjusted to magenta for users with red–green colour blindness. a, adhesive layer; r, resin tag. Scale bar, 10 µm. (*a*) Confocal image in the green channel showing the position of water within the dentinal tubules. (*b*) Red branch-like structures presenting adhesive permeation into the dentinal tubules without PAMAM-COOH pre-treatment. (*c*) Merged image of (*a,b*). (*d*) Confocal image in the green channel, showing the position of water within the dentinal tubules. (*e*) Red branch-like structures presenting adhesive permeation into the dentinal tubules with PAMAM-COOH pre-treatment. (*f*) Merged image of (*d,e*).

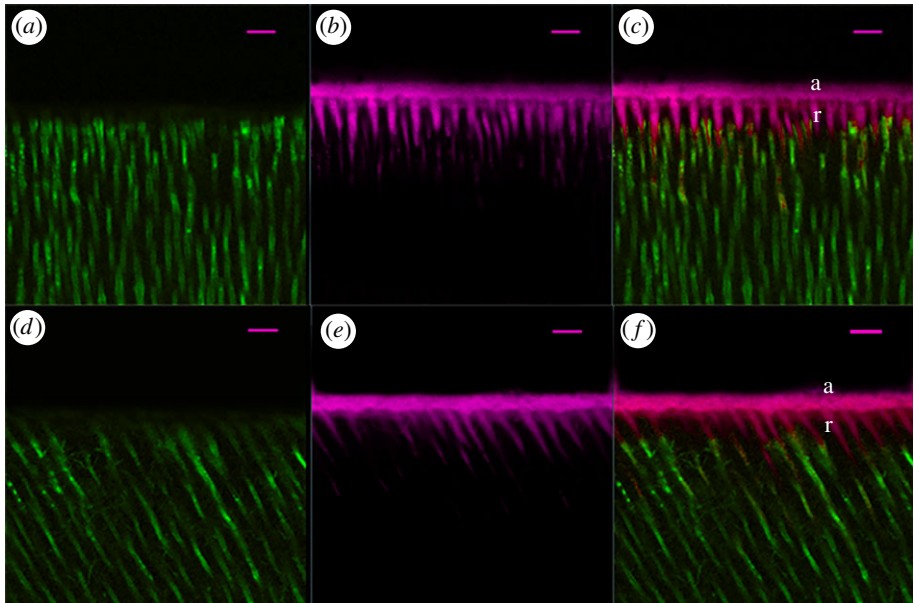

**Figure 9.** Representative CLSM fluorescence images of the resin–dentine interface created by the etch-and-rinse adhesive system under simulated pulpal pressure (5 cm of H$_2$O pressure) with or without PAMAM-COOH pre-treatment. The adhesive was dyed red, and water was dyed green. The red colour was adjusted to magenta for users with red–green colour blindness. a, adhesive layer; r, resin tag. Scale bar, 10 µm. (*a*) Confocal image in the green channel showing the position of water within the dentinal tubules. (*b*) Red droplet-like or short branch-like structures presenting adhesive permeation into the dentinal tubules without PAMAM-COOH pre-treatment. (*c*) Merged image of (*a,b*). (*d*) Confocal image in the green channel, showing the position of water within the dentinal tubules. (*e*) Red droplet-like or short branch-like structures presenting adhesive permeation into the dentinal tubules with PAMAM-COOH pre-treatment. (*f*) Merged image of (*d,e*).

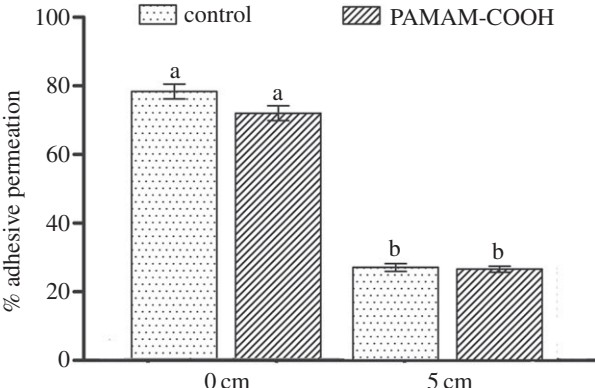

**Figure 10.** Mean and standard deviation of the relative percentage of resin adhesive permeation in the four subgroups. 0 cm, 0 cm of $H_2O$ pressure; 5 cm, 5 cm of $H_2O$ pressure; control, demineralized dentine interface pre-treated with deionized water; PAMAM-COOH, demineralized dentine interface conditioned with PAMAM-COOH. For the comparison of the four subgroups, the columns labelled with different lowercase letters are significantly different ($p < 0.05$).

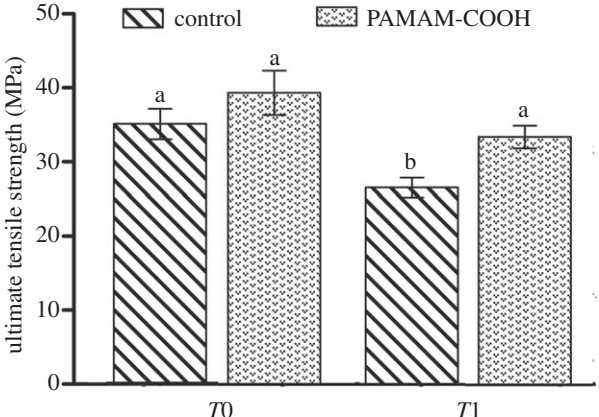

**Figure 11.** Mean and standard deviation of UTS in the four subgroups. $T0$, stored in deionized water for 1 day; $T1$, 10 000 thermal cycles; control, demineralized dentine interface pre-treated with deionized water; PAMAM-COOH, demineralized dentine interface conditioned with PAMAM-COOH. For comparison of the four subgroups, the columns labelled with the different lowercase letters are significantly different ($p < 0.05$).

between the relative permeation of the control group and that of the experimental group under the same simulated pulpal pressure ($p > 0.05$).

### 3.3.4. UTS testing

The UTS data are presented in figure 11. For immediate resin–dentine UTS, the experimental group showed no significant difference from the control group ($p = 0.643 > 0.05$). After thermal cycling, the value of UTS in the control group significantly decreased compared to the immediate UTS of the control group ($p = 0.002 < 0.05$). For the experimental groups, the UTS of the resin–dentine beam subjected to thermal cycling showed no significant decline ($p = 0.226 > 0.05$). In addition, the experimental group showed a significantly higher UTS than the control group after thermal cycling ($p = 0.022 < 0.05$).

## 4. Discussion

PAMAM-COOH is categorized as one type of hyperbranched polymeric macromolecules and has been reported to induce biomimetic remineralization of demineralized human tooth dentine [34,50,51]. According to the size-exclusion feature of collagen, G4-PAMAM-COOH could be incorporated within

collagen fibrils to guide hierarchical structure and intrafibrillar remineralization. Highly ordered biomimetic remineralization could protect the HL from collagen hydrolysis by endogenous MMPs and consequently increase the durability of the resin–dentine bond. However, achieving thorough remineralization of dentine collagen is time-consuming. During remineralization, the demineralized collagen matrix that serves as template for apatite mineral crystallite precipitation is inevitably subjected to degradation by endogenous dentine MMPs [3,52]. Notably, remineralization cannot occur in the absence of a collagen matrix. Therefore, our research aimed to explore whether G4-PAMAM-COOH can inhibit endogenous MMPs in the dentine matrix.

First, rhMMP9 as one representative type of dentine endogenous MMPs was selected to interact with G4-PAMAM-COOH *in vitro* by an rhMMP assay kit. Although the content of MMP-9 is inferior to MMP-2, the most abundant MMP in human teeth, they share a large number of constructional and functional similarities [13,53]. Furthermore, MMP-9, rather than MMP-2, can be additionally supplied from saliva [54]. The abundant saliva resources further contribute to the degradation of dentine matrix induced by MMP-9 [55]. In addition, there is a positive correlation between the quantity of MMP-9 in the dentinal fluid and the depth of the caries lesion, which clearly indicates that MMP-9 is a specific diagnostic indicator of dental caries [56,57]. Hence, exploring the effect of G4-PAMAM-COOH on soluble rhMMP9 becomes important for evaluating the ability of G4-PAMAM-COOH to inhibit the collagenolytic activities of the MMPs. The present study demonstrated that 1–10 mg ml$^{-1}$ G4-PAMAM-COOH could inactivate 90% soluble rhMMP9, which was comparable to the inhibitor provided by the MMP assay kits. Although this confirms that G4-PAMAM-COOH has an inhibitory effect on exogenous rhMMP9, its effect on endogenous MMPs enveloped inside mineralized dentine should also be investigated in view of the discrepant responsive environment. *In situ* zymography was performed to evaluate the proteolytic activity of the endogenous MMPs directly within the HL created by the etch-and-rinse adhesive system. In the present study, green fluorescence was detectable in four subgroups. Previous studies have suggested that demineralization by acid-etching could enhance proteolytic activity and adhesive application and further had an identical promoting effect [1,2]. This effect accounted for the visible green fluorescence in both the control and experimental groups. However, both observation by microscopy and quantitative analysis by ImageJ software represented that the fluorescence intensity was significantly different. Groups pre-treated with G4-PAMAM-COOH demonstrated weak proteolytic activity both initially and after thermal cycling, implying the inhibitory effect of G4-PAMAM-COOH on endogenous MMPs. Except for evaluating the enzymatic activities within the HL from a microcosmic perspective to verify the inhibitory effect of G4-PAMAM-COOH on endogenous MMPs, the predominant consequence of the interaction between endogenous MMPs and dentine collagen is the degradation of well-organized collagen into disordered fragments. Hence, collagen degradation by endogenous MMPs could be observed macroscopically by detecting the decomposed products of collagen. ICTP is a cross-linked carboxy-terminal telopeptide of type I collagen, resulting from collagen cleavage by MMPs. ICTP can be further decomposed by cathepsins to a short telopeptide called C-terminal cross-linked telopeptide (CTX). Notably, CTX would not be released from type I collagen in the absence of cathepsins. Therefore, the amount of ICTP in the incubation medium can indirectly reveal the collagenolytic activities of the MMPs. The present research detected the amount of ICTP released from the demineralized dentine powder after a two-week incubation. The release of ICTP in the experimental groups was significantly less than that of the control groups, suggesting that G4-PAMAM-COOH inhibited MMP-driven collagenolysis. In conclusion, the first null hypothesis that G4-PAMAM-COOH has no inhibitory effect on endogenous MMPs should be rejected.

Although the functional mechanism of interaction between PAMAM-COOH and MMPs was not explored, we surmise that the inhibitory effect of G4-PAMAM-COOH on MMPs is potentially related to its potency of chelation on Zn$^{2+}$ and Ca$^{2+}$. MMPs are classified as calcium- and zinc-dependent endoproteases. The zinc ion plays a vital role during the activation of an MMP, which is bound to the Zn$^{2+}$-active site of the catalytic domain. G4-PAMAM-COOH characterized as a multi-carboxyl dendrimer is theoretically able to chelate 32 Zn$^{2+}$ or 32 Ca$^{2+}$. A recent report confirmed that G3.5-PAMAM-COOH is a highly efficient Ca$^{2+}$ chelator equivalent to ethylene diamine tetraacetic acid (EDTA) [58]. In addition, PAMAM-COOH with a negatively charged surface radical might bind to proteins by electrostatic interactions, resulting in the inhibition of MMP collagenolytic activities. There have also been several studies concerning the interaction between PAMAM with different terminal groups and diverse proteins [28–30]. The enzymatic activity of a negatively charged protein (porcine pepsin) can be inhibited by PAMAM with a positively or neutrally charged surface. Nevertheless, PAMAM with a negatively charged surface has no impact on the same charged protein on account of electrostatic repulsion [30]. It is worth noting that proteins containing both positively and negatively charged regions might interact with

cation- and anion-terminated PAMAM dendrimers [59]. Hence, we speculate that the positive charge is dominated on the surface of MMPs or that there are positively charged areas in the structure of the MMPs resulting in the electrostatic interaction between G4-PAMAM-COOH and the MMPs.

The quality of the resin–dentine bonding is closely related to several aspects: the mechanical capacity of dentine collagen, the adhesive polymerization efficiency, the adhesive penetration of resin–dentine interface and the mechanical property of HL. In this research, G4-PAMAM-COOH was used as a primer for the purpose of satisfying the clinical application of resin–dentine bonding. Thus, the effect of G4-PAMAM-COOH pre-treatment on resin–dentine bonding was also necessary to be investigated.

It has been reported that cross-linking by agents such as glutaraldehyde and proanthocyanidins is an effective method to improve the mechanical property of dentine collagen [6,60,61] due to the generation of additional collagen cross-links by covalent and hydrogen bonds [9–11]. Hence, the present research intended to determine whether or not the molecular structure of dentine collagen changed after the G4-PAMAM-COOH pre-treatment and ascertain the mechanism of interaction between G4-PAMAM-COOH and collagen, which contributed to the evaluation of the effect of G4-PAMAM-COOH on the mechanical capacity of dentine collagen. FTIR spectra showed the frequency and intensity of the stretching vibrations and indirectly revealed the changes in the molecular structure of dentine collagen. In the FTIR spectra, G4-PAMAM-COOH pre-treatment did not obviously shift the frequency of each stretching vibration to a lower or higher wavelength (figure 6b,c), and the intensity ratio of the amide I band to the amide A band was not affected by G4-PAMAM-COOH. This demonstrated that the interaction between G4-PAMAM-COOH and collagen depended on neither the covalent bonds nor the hydrogen bonds and that dentine collagen was not cross-linked by G4-PAMAM-COOH. Hence, we conjectured that G4-PAMAM-COOH attached to collagen by electrostatic interaction. The results of adsorption and desorption verified this speculation. In figure 7, curve a is a total binding curve, representing the summation of G4-PAMAM-COOH bound to the collagen and dispersing into the interstitial water in the mineral-depleted region. The mineral-sparse space of dentine enlarged by demineralization procedures can be completely infiltrated with water. The minority of G4-PAMAM-COOH retained in the interstitial water was easily washed off by deionized water as shown in curve b. Curve c represents that the majority of the G4-PAMAM-COOH binding to collagen was extracted by sodium chloride, revealing indirectly that the predominant manner of interaction was electrostatic bonding. G4-PAMAM-COOH, categorized as an anionic dendrimer, might bind to the positively charged area of collagen. The chloride ions in the liquid are able to supersede the carboxylate anions to occupy the positively charged sites on the collagen. Hence, FTIR spectroscopy confirmed the results of G4-PAMAM-COOH absorption and desorption and determined that the exclusive manner of interaction between G4-PAMAM-COOH and dentine collagen was electrostatic adsorption rather than hydrogen or covalent bonding and that G4-PAMAM-COOH did not change the structure of dentine collagen, indirectly indicating that G4-PAMAM-COOH did not affect the mechanical property of dentine collagen. In addition, the elastic modulus of the demineralized dentine discs using AFM-based nanoindentation exactly affirmed the aforementioned deduction. Although the elastic modulus was not significantly increased, G4-PAMAM-COOH did not adversely affect the elastic modulus of the demineralized dentine matrix. Therefore, it was evident that G4-PAMAM-COOH did not damage the mechanical performance of dentine collagen exposed by acid etching during the process of resin–dentine bonding. The adhesive polymerization efficacy is another crucial factor for high-quality resin–dentine bonding [62,63]. In this work, the degree of conversion was assessed by micro-Raman spectroscopy, which directly reflect the polymerization efficacy. The results of DC demonstrated that G4-PAMAM-COOH did not decrease polymerization efficacy. Furthermore, adhesive infiltration into the dentinal tubules is equally important for strong resin–dentine bonding [64]. To evaluate the effect of G4-PAMAM-COOH on the adhesive permeation and morphology of the resin–dentine interface, the samples were observed by CLSM. To ensure the accuracy of quantitative evaluation on adhesive penetration, we selected three continuous areas of each specimen, located at the midpoint between the two enamel–dentine junctions, for gelatinolytic activity evaluation. The simulated pulpal pressures were designed to imitate the clinical oral condition. Although the simulated pulpal pressure deformed the shape of the resin tag from a branch type (in figure 8) into a droplet type (in figure 9), the resin tags at interface shared a morphological similarity in both experimental and control groups under 0 or 5 cm of $H_2O$ pressure. The quantitative analysis of permeability also exemplified that 8 mg ml$^{-1}$ G4-PAMAM-COOH as a primer did not decrease the permeability of the adhesive irrespective of whether 0 or 5 cm of $H_2O$ pressure was applied. Lastly, UTS was employed to evaluate the resin–dentine bond strength, revealing the mechanical property of HL. The results of UTS confirmed that pre-treatment of G4-PAMAM-COOH on dentine had no adverse effect on the immediate bond

strength. In addition, the application of G4-PAMAM-COOH improved the long-term bond strength. These findings were consistent with the outcomes obtained in the *in situ* zymography after thermal cycling. Our work verified that G4-PAMAM-COOH could dramatically inhibit endogenous dentine MMPs, which accounted for the improvement of the durability of the resin–dentine bond. In summary, G4-PAMAM-COOH has no negative impact on resin–dentine bonding, and so the second null hypothesis could be rejected.

Several studies have demonstrated that dendrimer toxicity is relevant to generation *in vitro* and *in vivo* [26,65]. Furthermore, at or below the fifth generation, there is no significant toxicity compared with at higher generations [26]. In addition, surface charge and terminal radicals play vital roles in dendrimer toxicity [65,66]. When intravenously and orally administered in mice, carboxyl-terminated dendrimers were preferably tolerated compared to amine-terminated dendrimers [67] and the anionic dendrimers were lysed less and were less cytotoxic than the cationic counterparts. It has been reported that a mouse fibroblast cell line manifested admissible cell viability when exposed to G4-PAMAM-COOH [68]. Therefore, it seems that G4-PAMAM-COOH is suitable for biological applications.

# 5. Conclusion

Within the scope of research restriction, it can be concluded that G4-PAMAM-COOH has an inhibitory effect on dentine endogenous MMPs and does not affect adversely properties of resin–dentine bonding. It is electrostatic bonding that assists G4-PAMAM-COOH attachment to collagen fibres. Results of this study not only support the conjecture that G4-PAMAM-COOH makes a contribution to protect dentine collagen matrix from collagenolytic activity by MMPs during remineralization, but also provide theoretical evidence for the further clinical application of G4-PAMAM-COOH in etch-and-rinse adhesive system.

Data accessibility. All data including electronic supplementary material of this study are available for anybody. Raw data used for plotting figures have been attached in the supplements. All figures and raw data have been submitted to the Royal Society Open Science website.
Authors' contributions. Q.W. contributed to design, acquisition of data, drafted and critically revised manuscript; T.S. contributed to analysis and interpretation of data, drafted the manuscript; M.Z. contributed to data acquisition and analysis, critically revised the manuscript; S.M. contributed to design, critically revised the manuscript; L.G. contributed to conception, design and data acquisition, drafted and critically revised the manuscript. All authors gave final approval and agree to be accountable for all aspects of the work.
Competing interests. We have no competing interests.
Funding. This work was supported by National Nature Science Foundation of China (grant no. 81873712) and Natural Science Foundation of Guangdong Province (grant no. 2018A030313409).

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
