## [Reviewer comments · Royal Society Open Science]

Review History

RSOS-182104.R0 (Original submission)

Review form: Reviewer 1

Is the manuscript scientifically sound in its present form?

Yes

Are the interpretations and conclusions justified by the results?

Yes

Is the language acceptable?

Yes

Is it clear how to access all supporting data?

Yes

Do you have any ethical concerns with this paper?

No

Have you any concerns about statistical analyses in this paper?

Yes

Recommendation?

Accept with minor revision (please list in comments)

Comments to the Author(s)

This study proposed a novel approaches to improve the durability of resin-dentine bonding. The authors attempted to ascertain the interaction mechanism between G4-PAMAM-COOH and dentine collagen. The study possesses a definite value of theory and guiding significance of application, but there are still some problems:

1. The authors need to answer very clear why is this PAMAM more important than the previous approaches to improve the durability of resin-dentine bonding?
2. In Fig 1, 2, 5, 10, 11, please mark the comparison group about the difference analysis.
3. Fig. 8, 9: no scale bar is found in the legend.
4. Avoid describing the results again in the discussion.
5. Where are the data of Nanoindentation by AFM and Degree of conversion (DC).
6. Data in table1 should be expressed as the final concentration of samples.
7. There are some grammatical errors in the text.

Review form: Reviewer 2

Is the manuscript scientifically sound in its present form?

Yes

Are the interpretations and conclusions justified by the results?

Yes

Is the language acceptable?

Yes

Is it clear how to access all supporting data?

Not Applicable

Do you have any ethical concerns with this paper?

No

Have you any concerns about statistical analyses in this paper?

No

Recommendation?

Accept as is

Comments to the Author(s)

The present study demonstrated that G4-PAMAM-COOH ranging from 1 mg/mL to 10 mg/mL inhibit 90% soluble rhMMP9. The release of ICTP in the experimental groups was significantly less than that of the control groups, suggesting that G4-PAMAM COOH inhibited MMP-driven collagenolysis. The present results clearly show that G4-PAMAN-COOH not only inhibited

exogenous soluble rhMMP9 but also hampered the proteolytic activities of dentine collagen-bound MMPs. Moreover, G4-PAMAM-COOH pretreatment did not significantly decrease the elastic modulus of the demineralized dentine, degree of conversion, penetration of the adhesive into the dentinal tubules or ultimate tensile strength. So G4-PAMAM-COOH treatment does not affect adversely properties of resin-dentine bonding. The results obtained are scientifically sound and original, as well as the conclusions of the work are based on the results.

Decision letter (RSOS-182104.R0)

14-Jun-2019

Dear Dr Gu,

The editors assigned to your paper ("The inhibitory effect of PAMAM-COOH dendrimers on dentine host-derived matrix metalloproteinases in vitro in an etch-and-rinse adhesive system") have now received comments from reviewers. We would like you to revise your paper in accordance with the referee and Associate Editor suggestions which can be found below (not including confidential reports to the Editor). Please note this decision does not guarantee eventual acceptance.

Please submit a copy of your revised paper before 07-Jul-2019. Please note that the revision deadline will expire at 00.00am on this date. If we do not hear from you within this time then it will be assumed that the paper has been withdrawn. In exceptional circumstances, extensions may be possible if agreed with the Editorial Office in advance. We do not allow multiple rounds of revision so we urge you to make every effort to fully address all of the comments at this stage. If deemed necessary by the Editors, your manuscript will be sent back to one or more of the original reviewers for assessment. If the original reviewers are not available, we may invite new reviewers.

- Data accessibility

<http://datadryad.org/submit?journalID=RSOS&manu=RSOS-182104>

- Competing interests

- Authors' contributions

- Acknowledgements

- Funding statement

Kind regards,

Andrew Dunn

Senior Publishing Editor

Associate Editor's comments:

Please respond fully in a point-by-point response document to the concerns of the reviewers. You should incorporate the changes they request in your manuscript document. Note that further consideration of your paper is contingent on satisfying both the editors and reviewers that your paper is ready for publication: only one round of revision is generally acceptable. As one of the reviewers comments that you need language editing support, please ensure you use a service such as those listed at <https://royalsociety.org/journals/authors/language-polishing/> to edit your paper before resubmitting.

Comments to Author:

Reviewers' Comments to Author:

Reviewer: 1

Comments to the Author(s)

This study proposed a novel approaches to improve the durability of resin-dentine bonding. The authors attempted to ascertain the interaction mechanism between G4-PAMAM-COOH and dentine collagen. The study possesses a definite value of theory and guiding significance of application, but there are still some problems:

1. The authors need to answer very clear why is this PAMAM more important than the previous approaches to improve the durability of resin-dentine bonding?
2. In Fig 1, 2, 5, 10, 11, please mark the comparison group about the difference analysis.
3. Fig. 8, 9: no scale bar is found in the legend.
4. Avoid describing the results again in the discussion.
5. Where are the data of Nanoindentation by AFM and Degree of conversion (DC).
6. Data in table1 should be expressed as the final concentration of samples.
7. There are some grammatical errors in the text.

Reviewer: 2

Comments to the Author(s)

The present study demonstrated that G4-PAMAM-COOH ranging from 1 mg/mL to 10 mg/mL inhibit 90% soluble rhMMP9. The release of ICTP in the experimental groups was significantly less than that of the control groups, suggesting that G4-PAMAM COOH inhibited MMP-driven collagenolysis. The present results clearly show that G4-PAMAN-COOH not only inhibited exogenous soluble rhMMP9 but also hampered the proteolytic activities of dentine collagen-bound MMPs. Moreover, G4-PAMAM-COOH pretreatment did not significantly decrease the elastic modulus of the demineralized dentine, degree of conversion, penetration of the adhesive into the dentinal tubules or ultimate tensile strength. So G4-PAMAM-COOH treatment does not affect adversely properties of resin-dentine bonding. The results obtained are scientifically sound and original, as well as the conclusions of the work are based on the results.

Author's Response to Decision Letter for (RSOS-182104.R0)

See Appendix A.

RSOS-182104.R1 (Revision)

Review form: Reviewer 1

Is the manuscript scientifically sound in its present form?

Yes

Are the interpretations and conclusions justified by the results?

Yes

Is the language acceptable?

Yes

Do you have any ethical concerns with this paper?

No

Have you any concerns about statistical analyses in this paper?

Yes

Recommendation?

Accept as is

Comments to the Author(s)

No

Review form: Reviewer 2

Is the manuscript scientifically sound in its present form?

Yes

Are the interpretations and conclusions justified by the results?

Yes

Is the language acceptable?

Yes

Do you have any ethical concerns with this paper?

No

Have you any concerns about statistical analyses in this paper?

No

Recommendation?

Accept as is

Comments to the Author(s)

The revised version of the manuscript is better and clearer than the previous one.

Review form: Reviewer 3

Is the manuscript scientifically sound in its present form?

Yes

Are the interpretations and conclusions justified by the results?

Yes

Is the language acceptable?

Yes

Do you have any ethical concerns with this paper?

No

Have you any concerns about statistical analyses in this paper?

No

Recommendation?

Accept as is

Comments to the Author(s)

In this manuscript, authors developed a novel approach to improve the durability of resin-dentine bonding. The study has a wide application in clinic. The study is novel and the manuscript is well written. I recommend this manuscript to be acceptance for publication in Royal Society Open Science.

Decision letter (RSOS-182104.R1)

08-Sep-2019

Dear Dr Gu,

I am pleased to inform you that your manuscript entitled "The inhibitory effect of PAMAM-COOH dendrimers on dentine host-derived matrix metalloproteinases in vitro in an etch-and-rinse adhesive system" is now accepted for publication in Royal Society Open Science.

Royal Society Open Science operates under a continuous publication model (<http://bit.ly/cpFAQ>). Your article will be published straight into the next open issue and this will be the final version of the paper. As such, it can be cited immediately by other researchers.

As the issue version of your paper will be the only version to be published I would advise you to check your proofs thoroughly as changes cannot be made once the paper is published.

Reviewer comments to Author:
Reviewer: 2

Comments to the Author(s)
The revised version of the manuscript is better and clearer than the previous one.

Reviewer: 1

Comments to the Author(s)
no

Reviewer: 3

Comments to the Author(s)
In this manuscript, authors developed a novel approach to improve the durability of resin-dentine bonding. The study has a wide application in clinic. The study is novel and the manuscript is well written. I recommend this manuscript to be acceptance for publication in Royal Society Open Science.

Appendix A

Response to Reviewer #1

This study proposed a novel approach to improve the durability of resin-dentine bonding. The authors attempted to ascertain the interaction mechanism between G4-PAMAM-COOH and dentine collagen. The study possesses a definite value of theory and guiding significance of application, but there are still some problems:

1. The authors need to answer very clear why is this PAMAM more important than the previous approaches to improve the durability of resin-dentine bonding?

Our response:

The application of MMP inhibitor and cross-linkers to improve the durability of resin-dentine bonding inevitably retain denuded collagen fibrils with poor mechanical properties. During the prolonged function, exposed collagen is susceptible to fatigue rupture and degradation. In the presence of non-collagenous proteins (NCPs), the biomimetic remineralization can restore the mechanical properties of flaccid collagen matrices beneath the HL to intact mineralized dentine characterized by excellent mechanical performance. Hence, biomimetic remineralization has been proposed as a method superior to the uses of cross-linkers and MMP inhibitors. Compared with the previous reported NCPs analogues such as polyacrylic acid and sodium tripolyphosphate, PAMAM-COOH is a multifunctional NCPs analogue, which can not only stabilize metastable amorphous calcium phosphate (ACP) precursors by their carboxyl groups but also orchestrate the alignment of ACP precursors during their transformation into apatite, resulting in mineralized collagen fibrils with hierarchy [24]. As the template for hydroxyapatite deposition, collagen scaffold plays an important role in biomimetic remineralization. Due to the extremely time-consuming process of in vivo remineralization, the susceptibility of collagen fibrils to degradation by active MMPs has become the main obstacle to obtaining satisfactory remineralization [35, 36]. Therefore, the present work examined the effect of fourth-generation PAMAM-COOH (G4-PAMAM-COOH) on the collagenolytic activities of endogenous MMPs and the durability of resin-dentine bonding subsequently.

The following sentences have been added to the Introduction (highlighted in red) in the

revised manuscript as follows:

“Similar to collagen cross-linkers, the main drawback of non-specific inhibitors to inactivate MMPs and cysteine cathepsins is the residue of the water-rich, resin-sparse and mineral-depleted collagen matrix within the HL. Although integrity of the denuded collagen fibrils can be preserved, the flaccid collagen matrices are susceptible to creep or cyclic fatigue rupture after prolonged function [18]. Deposition of apatite crystals in intrafibrillar and interfibrillar compartments is crucial for maintaining and stabilizing the mechanical properties of the dentine collagen [19]. Data reported in the literature suggested that biomimetic remineralization can restore water-filled, resin-sparse regions within the HL to intact mineralized dentine characterized by excellent mechanical properties and fossilize endogenous collagenolytic enzymes [20-22], resulting in enhanced resistance of collagen against fatigue rupture and degradation. Therefore, the biomimetic remineralization of demineralized dentine collagen beneath the HL has been proposed as a method superior to the uses of cross-linkers and MMP inhibitors. The implementation of biomimetic remineralization is closely dependent on three elements, involving non-collagenous proteins (NCPs), collagen scaffold and extraneous minerals (calcium and phosphorus).”

“NCPs are essential for the regulation of tissue mineralization by stabilizing metastable amorphous calcium phosphate (ACP) precursors and orchestrating the alignment of ACP precursors during their transformation into apatite [23, 24]. In view of the commercial unavailability of native or recombinant NCPs, scientists have resorted to the use of NCP analogues to mimic functional domains of naturally occurring proteins [23, 25]. Polyamidoamine dendrimers (PAMAM) are a new kind of hyperbranched polymeric macromolecules with well-defined dimensions and low cytotoxicity [26]. The structure of PAMAM can be classified into three components: the core, the interior and the shell [27]. The interior is composed of repetitive branching units, which dominantly affects the morphology of PAMAM. The more branching units it has, the higher generation PAMAM is defined as. The surface of PAMAM can be modified with functional peripheral radicals, introducing different surface charges. Several

studies have reported the ability of positively, neutrally and negatively charged dendrimers to interact with anionically and/or cationically charged proteins [28-30]. Furthermore, it has been reported that PAMAM possesses the properties of predominant biomimetic analogues [31-33], especially carboxyl-terminated PAMAM (PAMAM-COOH) dendrimers. Compared with the previous reported NCPs analogues such as polyacrylic acid and sodium tripolyphosphate, PAMAM-COOH as a multifunctional NCPs analogue can modulate highly ordered intrafibrillar mineralization on the organic dentine matrix [34]. Additionally, collagen scaffold, as the template for hydroxyapatite deposition, plays an important role in biomimetic remineralization. However, in vivo remineralization is an extremely time-consuming process; during remineralization, demineralized dentine collagen fibrils exposed by etching procedure may be susceptible to degradation by MMPs, resulting in unsatisfactory remineralization. Hence, the degradation of collagen fibrils by MMPs has become the main obstacle to obtaining satisfactory remineralization [35, 36].”

2. In Fig 1, 2, 5, 10, 11, please mark the comparison group about the difference analysis.

Our response: In the figure and table captions, the sentence “the columns labelled with the different lowercase letters are significantly different ($P < 0.05$)” has been added into revised manuscript and highlighted in red. And the figure 10 was corrected as follows in the revised manuscript.

3. Fig. 8, 9: no scale bar is found in the legend.

Our response: We thank the review for pointing this out, and scale bars (highlighted

in red) have been added into the legends of Fig 8 and Fig 9 in the revised manuscript.

“**Figure 8:** Representative CLSM fluorescence images of the resin-dentine interface created by the etch-and-rinse adhesive system without simulated pulpal pressure (0 cm of H₂O pressure) with or without PAMAM-COOH pre-treatment. The adhesive was dyed red, and water was dyed green. a, adhesive layer; r, resin tag. **Scale bar, 10 μm.**”

“**Figure 9:** Representative CLSM fluorescence images of the resin-dentine interface created by the etch-and-rinse adhesive system under simulated pulpal pressure (5 cm of H₂O pressure) with or without PAMAM-COOH pre-treatment. The adhesive was dyed red, and water was dyed green. a, adhesive layer; r, resin tag. **Scale bar, 10 μm.**”

4. Avoid describing the results again in the discussion.

Our response: We have made modifications (highlighted in red) in the revised manuscript to be more concise.

5. Where are the data of Nanoindentation by AFM and Degree of conversion (DC).

Our response: The data of Nanoindentation by AFM and Degree of conversion (DC) have been shown in the Results session of the revised manuscript as follows:

4.3.1 Nanoindentation by AFM

The elastic modulus of the demineralized dentine discs immersed in deionized water and 8 mg/mL G4-PAMAM-COOH were **4.06±0.45 MPa and 4.72±0.57 MPa**, respectively. Statistical analysis showed that the elastic modulus of the dentine discs in the two groups was not significantly different ($P=0.425>0.05$).

4.3.2 Degree of conversion (DC)

In the control group, the value of DC was **70.95±0.90%**; the value of DC in the experimental group reached **69.03±1.80%** and was not significantly different from that of the control group ($P=0.362>0.05$).”

6. Data in table1 should be expressed as the final concentration of samples.

Our response: Final concentrations of samples have been added into the Table 1 as

follows:

Group	Test solution and volume	Final concentration of G4-PAMAM-COOH
1	1000 μ L deionized water	0mg/mL
2	500 μ L artificial saliva, 500 μ L deionized water	0mg/mL
3	500 μ L 16 mg/mL G4-PAMAM-COOH (dissolved in deionized water) 500 μ L deionized water	8mg/mL
4	500 μ L 16 mg/mL G4-PAMAM-COOH (dissolved in deionized water) 500 μ L artificial saliva	8mg/mL

7. There are some grammatical errors in the text.

Our response: We have carefully proof read the manuscript and made modifications (highlighted in red) in the revised manuscript.